# Co-design and evaluation of a youth-informed organisational tool to enhance trauma-informed practices in the UK public sector: a study protocol

Siobhan Hugh-Jones [1], Isabelle Butcher [2], Kamaldeep Bhui [3]

¹Psychology, University of Leeds Faculty of Biological Sciences, Leeds, UK
²Department of Psychiatry, University of Oxford, Oxford, UK
³Department of Psychiatry and Nuffield Department of Primary Care Health Sciences, University of Oxford, Oxford, UK

**Correspondence to**
Professor Siobhan Hugh-Jones;
s.hugh-jones@leeds.ac.uk

## ABSTRACT

**Introduction** A trauma-informed approach (TIA) means working with awareness that people's histories of trauma may shape the way they engage with services, organisations or institutions. Young people with adverse childhood experiences may be at risk of retraumatisation by organisational practices in schools and universities and by employers and health agencies when they seek support. There are limited evidence-based resources to help people working in the public sector to work with adolescents in trauma-informed ways and the needs of adolescents have not been central in resource development. This study contributes to public sector capacity to work in trauma-informed ways with adolescents by codesigning and evaluating the implementation of a youth-informed organisational resource.

**Methods and analysis** This is an Accelerated Experience-based Co-design (AEBCD) Study followed by pre–post evaluation. Public sector organisations or services, and adolescents connected with them, will collaboratively reflect on lived experience data assembled through creative arts practice, alongside data from epidemiological national data sets. These will present knowledge about the impact of adverse childhood experiences on adolescents' mental health (stage 1). Collaboratively, priorities (touch points) for organisational responses will be identified (stage 2), and a low-burden resource will be codesigned (stage 3) and offered for implementation (stage 4) and evaluation (stage 5) in diverse settings. The study will provide insights into what adolescents and public sector organisations in the UK want from a TIA resource, the experience of services/organisations in implementing this and recommendations for resource development and implementation.

**Ethics and dissemination** The UK National Health Service Health Research Authority approved this study (23/WM/0105). Learning will be shared across study participants in a workshop at the end of the study. Knowledge products will include a website detailing the created resource and a youth-created film documenting the study process, the elements of the codesigned resource and experiences of implementation. Dissemination will target academic, healthcare, education, social care, third sector and local government settings via knowledge exchange events, social media, accessible briefings, conference presentations and publications.

## STRENGTHS AND LIMITATIONS OF THIS STUDY

⇒ This is a multisite study nested study in a larger national study drawing on youth lived experience and survey data to inform the codesign of a resource to support diverse public sector settings to work in trauma-informed ways.

⇒ We plan to recruit diverse young people to work in partnership with the organisations through which they are recruited, so that youth voice informs the codesign process. Our study also aims to understand how to deliver coproduction studies with young people in ways that are trauma informed.

⇒ Use of frameworks for experience-based codesign and implementation theory is a strength.

⇒ Implementation plans and evaluation protocols will be codesigned with settings, improving the chances of acceptable and meaningful resource use and evaluation.

⇒ A limitation is that the time for implementation is relatively short, and not formalised, relying on participants adopting the interventions in their workplaces and evaluation stage is a simple real-world pre–post assessment, not a trial.

## INTRODUCTION

Adverse childhood experiences (ACEs) include events such as verbal, sexual or physical abuse; neglect; parental separation or incarceration; substance misuse, domestic violence or family mental illness, bullying, poverty, racism, death of significant others and multiple losses.[1 2] Children who have experienced ACEs are at elevated risk of poor mental health outcomes compared with children who have not experienced one of these ACE.[3] Experiencing multiple ACEs can have particularly detrimental effects on a physical, mental, social and economic outcomes in childhood, adolescence and adulthood.[4 5]

Many ACEs meet the definition of trauma[6] as events that involve actual or threatened death, serious injury or sexual violence either by witnessing it or hearing about it.[7] Although many ACEs (eg, homelessness) are

not included in this definition of trauma, many ACEs have trauma effects.[1] In our work, ACEs are considered as potentially traumatising, especially in the absence of buffering, supportive relationships and contexts. Our focus is the long-term impact of ACEs on later adolescent mental health.

Trauma can powerfully shape a young person's psychobiology,[8] influencing their mood, thoughts, emotions and behaviours in adolescence as well as how they view, make sense of and react to events and people (eg, being on alert for harm from people, anticipating negative outcomes). Such responses are adaptive following trauma, and dynamic in that responses to new traumas may change. However, usually the style of adaptation tends to persist even when a trauma has stopped. The response can overgeneralise into non-trauma situations (Lis *et al*) as threat perceptions are altered and sensitised.

Adaptive responses can include avoiding specific reminders, loud noises, too many people, or the need to calm and self-soothe at a time of anxiety and autonomic arousal linked to the flight-fight response. Alongside these, there may be emotional dysregulation and impulsivity and inability to plan if a person is affected by flashbacks and distress, as if the original event is being repeated. Therefore, a person's 'adaptive' behaviour and their responses can easily be misunderstood or misrepresented by others as unhelpful, disproportionate, confusing or defensive or even showing poor conduct and unprofessional. This may result in actions by the teacher or employer, for example, to apply rules and codes of conduct, and sanctions.

These can then be felt as a form of retraumatisation; this occurs when experiences that are mundane to others (eg, meeting professionals, being asked to consent to a process) remind the person of their past trauma, and a trauma-based response is evoked (eg, submission, disengagement, fear-based aggression).[9] Many community organisations and services, from schools to youth organisations and healthcare, routinely encounter young people who may have experienced ACEs but may be unaware of how the young person's trauma may be shaping their needs, engagement and behaviour with them.[10 11]

Since the early 2000s, there has been increasing global attention to the ways that organisations, services, institutions and settings (hereafter organisations) could work in trauma-informed ways.[12 13] Trauma-informed approaches (TIA) aim to improve the capacity of professionals and services to respond helpfully to people who may have experienced trauma. TIAs, although still being defined, concur that the aim is not to treat trauma-related difficulties, which is the remit of trauma-specific services. Rather, TIAs focus on people and processes and seek to improve organisational understanding that trauma is often inflicted by someone previously trusted by a young person and to appreciate the prevalence and impact of trauma on a person's neurological, biological, psychological and social development as well as how surviving adversity may be shaping a person's coping strategies.[14 15] TIAs align with a social model of disability, which views

societal lack of adaptation and flexibility to be generating disability rather it being an innate property of the person. The key principles of a TIA, which have broad international recognition, are safety, trustworthiness, choice, collaboration and empowerment.[12 16] Yatchmenoff *et al*[14] summarise these as safety, empowerment and self-worth.

In the UK, the government has produced guidance on TIAs for the health and social care sectors.[17] The guidance presents TIAs as understanding how the trauma-imprint can affect a person's ability to develop trusting relationships with organisations and staff alongside the importance of minimising the risk of retraumatisation (which can undermine recovery) and of increasing a sense of control and autonomy for the trauma survivor.[12] The way organisations function, their policies and procedures and how they engage with and respond to people can all carry retraumatisation risks.[18] The guidance also acknowledges that trauma, both direct and vicarious, can affect staff and thus protecting their well-being is also a feature of a TIA.[16]

TIAs are widely recommended, and often implemented, across public sectors in the UK,[16 19 20] including healthcare, education and social services. Although the evidence of effectiveness is only emergent and is highly complex to gather, TIAs appear to have potential to be beneficial for ACE-affected populations.[21–23] However, despite an organisation's commitment to ideology of a TIA, they can, like any complex intervention, be hard to implement and sustain.[24 25] They require change at a systemic level to support paradigm shifts across policy, procedures and practices, finding ways to in trauma-informed ways in every interaction and to prioritise the building of trusting, mutual relationships.[26] Organisations need support to become TI.

Multiple resources and toolkits have been developed internationally to assist organisations in implementing a TIA.[27 28] Toolkits have begun to emerge in the UK. For example, the Roots framework[26] is a self-assessment tool facilitating organisational reflection on current and possible practice, including in National Health Service settings. The Scottish government has produced a Trauma-Informed Practice Toolkit for sector-wide organisations to self-evaluate and plan action across 10 implementation domains.[29] However, despite a proliferation of resources for TIAs, young people's perspectives in shaping these remain rare. There is also a need for continued learning on successful implementation of TIAs, to see what TIAs look like in practice in different human service organisations, the barriers encountered and how these can be overcome.[14] A further significant gap in knowledge is the impact of TIAs. Existing evidence comes mostly from clinical settings, where professionals are likely to have specialist trauma training. There has been little evaluation of TIAs in UK public sector organisations whose primary focus is not clinical care.

Our *Attuned to Trauma* study aims to contribute to filling these knowledge gaps. We present our protocol for this study, which is nested within a large, UK project (*Attune*)

examining ACEs and pathways to mental health outcomes in adolescents. Our nested study builds on the learning from two early work streams in Attune, spanning arts-based lived experience and data set analyses. Combining insights from those work streams and using AEBCD principles,[30] we bring young people and public sector organisations together to codesign, implement and evaluate a new youth-informed resource to support TIA in organisations relevant to adolescents (10–24 years) in the UK. We use this age range as it is encouraged by the project funders (UK Research and Innovation) and reflects calls to appreciate the earlier onset and extension of this life phase, arguably driven by social, digital, marketing, education and economic forces (Sawyer *et al*). Working with young people, via participatory and codesign research methods, can increase the chances of service improvements, which are sensitised to their needs and lived experience, and, therefore, more likely to be effective. Our project aligns with coproduction theory that 'service users' should not be seen as passive recipients of that service but as active citizens with something of value to contribute. Treating young people's perspectives as an equitable form of knowledge is a core theme of the UKRI funding stream for this project. We, therefore, bring professionals and young people into dialogue and cocreation together.

### Study aim

This study aims to codesign, implement and evaluate a public health resource to help diverse public sector organisations in the UK to work in TI ways with adolescents and young people (10–24 years). Our study also aims to contribute to understanding about how to deliver coproduction studies with young people in ways that are trauma-sensitive.

### Research questions

We address two primary research questions:
1. What do young people and stakeholders want in a codesigned UK resource to improve public sector capacity to work in trauma-informed ways?
2. Is such a codesigned resource acceptable, feasible, efficacious and affordable, and how should it be refined for future implementation in the UK public sector?

## METHODS

### Design

This is a codesign and pre–post evaluation study nested within the UK Attune project. The codesign stage draws on experience-based co-design (EBCD), which is widely used to design health service improvements. Central to EBCD is an initial discovery phase, which brings together service users and service providers in a balanced power relationship to discuss the personal experiences of service users and key moments in the service that shaped this. Service users and providers are then supported to codesign service changes to improve the quality of care.[31] We will adopt an accelerated form of EBCD, termed AEBCD, which can be effective in achieving similar aims to standard EBCD. AEBCD typically brings pre-existing narratives of experiences from relevant groups to start the codesign process, establishing where the codesign group identifies consensus or divergence from those narratives. We will do this by bringing early Attune insights from adolescents (instead of the usual service user interviews), and adolescent-produced art, which includes film, and narratives to convey lived experience (instead of a purposefully commissioned service user 'trigger' film). We retain the core principle of EBCD in that the experiences of young people will drive codesign of organisational resource(s) for a TIA. The produced resource(s) will be shared with specific organisations to explore how they chose to implement it. We will conduct a preliminary evaluation of this.

Normalisation process theory (NPT) and principles of diffusion of innovation[32] in service organisations will inform our approach to resource coproduction, trial implementation and evaluation, acknowledging we are at the earliest stage of innovation for service development. NPT characterises implementation as a social process of collective action. It proposes concentration on four domains at all stages of complex intervention design, delivery and evaluation, namely intervention coherence, cognitive participation (engagement), collective action (required to enable the intervention to happen) and reflexive monitoring (of the costs and benefits of the innovation). Attention to these domains will structure our project approach. We will supplement these with the most relevant domains influencing the diffusion of innovations in services and organisations, for example, characteristics of the system, the innovation and anticipated individual adopters.[32 33]

### Setting

This is a multisite study taking place across three regions of England, namely Cornwall, Kent and West Yorkshire. These sites are chosen to support participation by young people and stakeholders from rural, coastal and urban regions. Place is important in understanding ACEs[34–36] and in understanding local organisational cultures and resources for change.

### Participants

#### AEBCD stage

There is little literature to inform trauma-informed, codesign methods (see McGeown *et al* and Cherry[37 38], eg, in primary healthcare intervention codesign), although principles of codesign overlap with some principles of a TIA (eg, collaboration, choice). We aim to contribute to emerging discussions about how to deliver trauma-informed coproduction methods by recruiting five young people from the Attune regional young people's advisory groups (YPAGs) to form a working party to design the AEBCD workshops to be trauma-sensitive. The YPAGs are based in several regions in the UK: Cornwall, Kent, London, Oxfordshire and Yorkshire. The YPAGs comprise young people between 10 and 24 years with diverse

identities (eg, in terms of neurodivergence, gender identity, ethnicity and gender).

Sample sizes for EBCD studies vary. The recommended number of service users is between 5 and 15. For our AEBCD workshops, we will aim to recruit 5–8 young people (10–24 years) and between 5 and 8 professionals to attend workshops in each of our UK regions (Cornwall, West Yorkshire and Kent; highest total n=48). Codesign processes need to be responsible to participants with whom power is shared; therefore, despite this intended plan, we will retain flexibility to respond to the needs of young people and stakeholders who may prefer session in smaller or larger groups, or in a modified format.

Professionals will be recruited from diverse public sector organisations in our nominated regions. This may be through the Attune project network or by first approaches to local councils, local authorities and/or third sector organisations. We will seek organisations who have a helping role with young people, for example, schools, local authorities, colleges and universities, youth offending teams, community support organisations, housing associations, etc. Bringing diverse organisations together to work on TIAs can be effective when they are supported to focus on their similarities, common challenges and collective solutions rather than differences.[14] Working in local, regional contexts can support interprofessional learning.[39] To be eligible, organisations must confirm they are ready and motivated towards TI ways of working and commit to sending up to two organisational members of staff to all AEBCD workshops and to supporting the recruitment of two to three young people from their setting with whom they can work in partnership.

Young people will be recruited via their participating organisation/setting. Participating organisations will be asked to identify 2–3 young people from their setting who are between 10 and 24 years; able to give consent and, if under 16, provide guardian consent; have experienced trauma and/or ACEs; and able to contribute to AEBCD workshops (conducted in English) where ACEs and trauma will be discussed. Those recruited young people will have experienced trauma and/or ACEs will be determined via three means: (1) knowledge that the recruiting setting has about the young person whom they deem suitable to approach for study participation; (2) our recruitment information to young people will include experience of trauma/ACEs as an eligibility criterion and (3) via a survey administered as part of the larger Attune survey that includes items about experiences of ACEs and trauma. In considering potential participants, organisations will be asked to consider approaching underrepresented groups where possible (ie, young people from minority communities, with diverse identities in terms of place, gender, ethnicity, Lesbian, Gay, Bisexual, Transgender, Queer, Intersex, Asexual, including Pansexual and Two-Spirit (LGBTQ+) and neurodivergence).

Organisations will be asked to approach potential participants with study information and explore their interest in taking part and to secure informed consent from them and their guardians as needed. All young people will be paid for their time in line with national guidance.[40] Requiring guardian consent for under 16s, which is mandated by the approving ethics committee, may prohibit some young people from participating (eg, those still living in adverse environments). Participation by professionals and young people is opt-in.

## Patient and public involvement
Patients and/or the public were involved in the design, or conduct, or reporting, or dissemination plans of this research.

## Implementation and evaluation
All organisations who participated in the AEBCD across our study regions will be eligible for the next stage of implementing and evaluating the co-designed resource for 6 months.

## Procedure
### AEBCD stage
#### Pre-workshop preparation (summer 2023)
Taking part in coproduction for TIAs may generate individual and organisational concerns, for example, around one's own trauma, managing personal and organisational disclosures, or about organisational limits for change. As part of our efforts to design trauma-informed coproduction methods, we will prioritise the fostering of psychological safety, a first principle of a TIA, for organisation staff. Applied to working practices, psychological safety is seen as essential to 'unfreezing' organisations, so they can learn and change[41] and is manifested by 'the willing contribution of ideas and actions to a shared enterprise'[42](p24). It can be fostered by reducing perceived threats and interpersonal risk and encouraging provisional tries towards change.[43] Therefore, using a topic guide informed by NPT and Greenhalgh et al's[32 33] work on developing and implementing complex interventions, we will conduct approximately three individual online preworkshop orientation meetings with organisations to help us understand their reasons for taking part, their needs and expectations of the study, the main demographic of young people they encounter, their current status with regards to TIAs, any strategic objectives informing their current or planned organisational delivery or organisational practice, their perceived risks around participation and how we can mitigate these (ie, system antecedents and readiness for innovation).

We will also endeavour to promote psychological safety for adolescent attendees. For adolescents, psychological safety can means feeling able to take interpersonal risks because there is little fear that this will result in embarrassment, ridicule or shame and so it enables people to engage, connect, change and learn.[42] Positive relationships, allyship through working with peers similar to you and control over contributions can all build psychologically safe working contexts for adolescents.[44] Towards

this, we will conduct online engagement meetings with adolescents to begin to build positive relationships, to understand their motivations, needs, expectations and concerns about participation and how we can mitigate any perceived risk, and optimise their ability to contribute. Adolescent attendees will also be invited to meet each other online before the first workshop and to identify shared ambitions for their involvement.

With consent, and with the help of the youth working party, appropriate information from the online premeetings with organisations and young people (eg, strategic objectives of the organisation, perspectives on how to contribute) will be synthesised into a collaborator's booklet, so that organisations and young people are aware in advance of who they will be working with in the workshops and how people can contribute and benefit differently.

Signed informed consent for the ECBD workshops will be secured after these premeetings from all confirmed attendees. Our learning from these meetings will shape workshop design. We will also secure opt-in or opt-out preference for photography and filming in the workshops.

### AEBCD workshop series
We will conduct four full-day AEBCD workshops over 18 months; three before the evaluation stage and one after for optimisation and lessons learnt from the evaluation. Table 1 shows the aims and anticipated outcomes of each workshop and figure 1 shows the study timeline and connection with the main Attune project. Young people will lead many workshop sessions and workshop facilitators will be members of the Attune research team who will be trained and follow a standardised format for each workshop.

Workshops will involve activities and discussion in small and large groups. Discussion questions will be shaped by the youth working party, so they are framed in ways that are meaningful and accessible to adolescent attendees; as far as possible, planned questions and activities will be sent to all attendees in advance of workshops. Conventions in EBCD are to use basic but effective means of securing participants' engagement (eg, creative activities, scenarios, case discussion) and responses (eg, flip charts, post-it notes, maps, sketches, etc). We offer examples below of the types of activities we may use and additional detail is given in online supplemental materials. We will emphasise our valuing of multiple ways of contributing beyond talking (including listening, responding to anonymised digitised questions such as Mentimeter, postevent feedback) from which we can learn. These are features of psychologically safe ways of working with young people. We will seek brief workshop evaluation via email after each workshop.

### Data collection
AEBCD workshops will be audio-recorded (small group and/or whole group level) and field notes will be taken by trained facilitators. We will collect all workshop artefacts

(eg, creative outputs, written and digital responses) to support analysis. As soon as possible after each workshop, facilitators will meet to share notes and to reach consensus on key insights from the events. At the end of the AEBCD stage, we will evaluate our approach using the Public and Patient Engagement Evaluation Tool, which assesses the quality and impact of public and patient engagement activities within health system organisations.[45]

### Data analysis
The key moments in the workshop audio-recordings will be transcribed (ie, those which are pertinent to learning, excluding for example, team building exercises). Transcribed extracts will only be linked to organisation type when essential, for example, when knowledge of the setting type is required to comprehend the point/discussion. Full anonymisation of any other identifying details will occur at the point of transcription. Analysis will be conducted at regional levels before being merged for whole group analysis. We will merge the 'key insights' field notes with workshop artefacts and 'key moment' in the audio recording, initially framed in terms of workshop objectives (eg, to understand views on an issue or to identify priorities). We will share emerging outcomes with the wider Attune project for credibility and sense-checking at key points. Lay summaries of each workshop will also be shared with participants prior to the next workshop, where they will be invited to approve or improve our recording and interpretation of learning from the preceding workshop. Data storage and sharing will be in line with the data management plan for the Attune project.

### Workshop 1 (autumn 2023)
The EBCD discovery phase typically involves understanding the organisation and 'service users' lived experiences. Workshop 1 is a discovery phase where we share Attune Workstream 1 data on the lived experiences of ACE-affected young people, generated using arts-based approaches, their views on how this has affected their mental health and how the insights are relevant for organisations. We will also present insights from Attune Workstream 2, which involves the analysis of several existing, large UK data sets to identify determinants of adolescent mental health risk and resilience following ACEs. Discussion will then explore key touchpoints,[46] namely (1) how these insights about the imprint of trauma on adolescents lead organisations to reflect on and identify some prominent current practices, which may be unattuned to this knowledge and (2) what kinds of help organisations feel would help them to become more youth-informed and trauma-informed. We avoid introducing existing tools at this point to ensure service and service-user led solution.

### Workshop 1→ workshop 2
In line with our data analysis plan, we will examine how the discovery phase impacted different sectors/organisations and where they identified priorities and opportunities for change. Our analysis will be attentive to

**Table 1** Aims and anticipated outcomes from AEBCD workshops and informing concepts

| Workshop | Primary aim | Secondary aim | Outcomes | Informing concepts |
|---|---|---|---|---|
| 1 | To share lived experience and data insights from and about young people's ACEs and trauma (from Attune) to prompt reflection on organisational change. | Establish collaboration, model a trauma-informed way of working, promote youth voice, understand the readiness and needs of organisations from youth and professional perspectives to act on new knowledge. | Learning what the data means to diverse sectors and where organisations recognise need, motivation and opportunity to operationalise a TIA. | Linkage* Communication & Influence* Absorptive capacity for new knowledge* System readiness* |
| Methods | Presenting Attune artworks, films, key themes from Attune WS1 (lived experience extracts) and WS2 (outcomes of national dataset analyses). Enquiry activities about participants' views of this, the level of resonance/divergence and what it directs attention to for public sector settings. | Pre-workshops communications and meetings to understand and build readiness and collaborative intention; being clear about the day's structure, content and purpose; dedicating safe spaces in the workshop and a trauma-informed facilitator to support young people; careful orientation (ice-breakers) opportunities; workshop activities based on play and arts (suitable for all ages) and multiple ways of contributing (drawing, writing, hand signals). | | |

**After workshop 1:** Synthesise outcomes across regions ready for sharing at workshop 2.
If appropriate, share existing organisational TI principles and toolkits as stimulus.

| Workshop | Primary aim | Secondary aim | Outcomes | Informing concepts |
|---|---|---|---|---|
| 2 | To identify priority, modifiable ways of working to become trauma-informed and identify the nature of a resource to be created. | To strengthen partnerships, to understand where youth and organisational perspectives on the nature of a TIA align and where they do not; and to understand the values and motivations of organisations for change, and what they need from a resource to help them do this, some expected outcomes for evaluation | Learning what organisations need from a resource, the most modifiable aspects of their ways of working, and likely benefit to them and young people. | Linkage* The innovation* Adopter* System readiness* |
| Methods | Exploration of a select number of existing tools; imagination tasks for co-design; consensus-reaching activities (nominal group technique; bullseye tasks); co-design activities | Invigorating (through words) project ambition and potential through partnership; relationship building activities; activities exploring differences and similarities in adolescent and professionals' priorities; play building activities to explore opportunities and challenges in co-design | | |

**After workshop 2:** Synthesise outcomes and share.
Produce toolkit prototypes, and draft implementation and evaluation plans

| Workshop | Primary aim | Secondary aim | Outcomes | Informing concepts |
|---|---|---|---|---|
| 3 | To select and refine core components of the final resource prototype | Decide on implementation options for the next phase and establish priorities for the evaluation. | Prototype resources and recommendations for implementation and evaluation. | Linkage* Implementation process* |
| Methods | Goal agreement; nominal group technique; priority rating activities for outcomes | Activities to explore aspects of NPT with professionals and (separately) to explore pros and cons of implementation options with young people; implementation roadmap design | | |

**After workshop 3:** Refine prototype resource, implementation plan and evaluation protocol.

Continued

| Workshop | Primary aim | Secondary aim | Outcomes | Informing concepts |
|---|---|---|---|---|
| 4 | To review findings from the implementation and evaluation phase and improvements needed to the resource | To capture implementation experiences and recommendations for an improved implementation protocol; to understand lessons learnt and ambitions of youth and organisation for TIA in the future | Understanding how to improve the resource, and its implementation and evaluation, ready for future testing. | NPT domains of evaluation: coherence, cognitive participation, collective action, reflexive monitoring† |
| Methods | Arts-based activities to capture experience; case discussions; comparative outcomes to expectations | Priority rankings for learning and development; creating key message banners | | |

*From Greenhalgh *et al*[32] conceptual model of the determinants of diffusion, dissemination and implementation of innovations in health service delivery and organisation.
†NPT (Murray *et al* [48]).

organisation type if necessary (ie, if differences emerge). Outcomes will be summarised by region first and then combined with outcomes from the other study regions, where appropriate. Regional and merged outcomes will be shared with regional attendees who will be encouraged to share and discuss this in their organisation.

### Workshop 2 (spring 2024)

As stimuli for workshop 2, and if workshop 1 data indicate relevance and need, we will share in advance with participants two existing UK toolkits for organisations to become more trauma-informed.[26] These toolkits are recent and evolving but are not yet youth-informed and are resource-intensive to implement. In workshop 2, we will begin codesign of a low-intensity, youth-informed organisational resource. The aims of this workshop are to (1) identify modifiable ways of working to become trauma-informed and which are highly likely to be of benefit to adolescents; (2) likely to be adopted (as per Greenhalgh *et al*[32] characteristics of the innovation and system) and (3) begin to develop the nature of resource to be created. If consensus from codesign is lacking across our regional sites, we will convene a small working group with representation from each site to produce a final set of priorities.[47] These will especially focus on specifications that permit the resources to be adopted for particular sectors, regions, and intersectional characteristics encountered in stakeholder organisations.

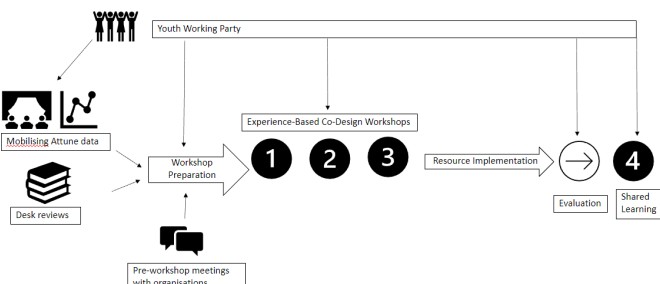

**Figure 1** The timeline of the study and connection to wider Attune Project.

### Workshop 2 → workshop 3

Once outputs are analysed, the research team and youth working party will create up to four outline prototype resources that satisfy some of the needs expressed in the workshops. We will also produce draft implementation and evaluation recommendations, all shared in advance of workshop 2, so that organisations have opportunities to discuss at a local level.

### Workshop 3 (Easter 2024)

The aim of this workshop is to select and build the core components of a final resource as well as final recommendations for its implementation and evaluation. Workshop outputs will inform development, with the youth working party, of a final resource ready for the implementation and evaluation phase.

### Workshop 4 (spring 2025)

After the implementation and evaluation phase, workshop 4 will bring participants back together to (1) share group-level analysis of outcomes and experience of implementation and (2) key aspects of the resource to refine for testing in future studies. We will capture organisational plans to continue with any TIA changes as well as youth and professional evaluations of study participation, including psychological safety.

### Implementation and evaluation

Implementation of new innovations can be optimised by the involvement stakeholders in the earliest stages of design and ownership over local and flexible implementation.[32 48] Organisations involved in our AEBCD stage will be invited to implement the resource in their setting for a minimum of 6 months. Although AEBCD will produce implementation recommendations, organisations will have autonomy in adapting the resource and its implementation to their setting, reflecting a supported 'let it happen' approach to learn from organisational agency and creativity in implementation.[49] This is in line with moves away from prioritising intervention fidelity over local tailoring, especially in the learning phase of

interventions.[50] As a minimum, and in line with recommendations for TIA, organisations will be encouraged to establish an implementation team to lead and champion resource implementation in their setting.[14]

Evaluation of the impact of TIA on service users is extremely complex[23] and is not the focus of our early stage evaluation. We will evaluate resource implementation. Administering a standardised pre–post measure of an organisation's TIA may not be feasible as there are no UK generated measures suitable for the evaluation of TIA in diverse settings such as those likely to be involved in our study. Thirkle *et al*'s review[51] of measures of TIAs shows that most tools originated in the USA and are not easily tailorable beyond healthcare organisations.[12] Decisions on whether to use a standardised measure or to generate a bespoke pre–post evaluation tool will be made in the AEBCD stage. Evaluation will be informed by NPT and relevant domains of diffusion of innovations in services.[32 33 48] It will include, as a minimum, a survey of how organisations deployed the resource and audio-recorded, key informant semistructured interviews, following a topic guide, to explore the experiences of organisations; barriers and solutions to resource use; any unanticipated benefits or adverse impacts; lessons learnt and recommendation for improvements, testing and upscaling. We will also consider the extent to which the NPT parameters for optimal implementation were met (coherence, cognitive participation, collective action and reflexive monitoring). As in all intervention research, it is important to learn if outcomes, especially poor ones, were influenced by the resource itself or by challenges in implementation.[52] Qualitative data will be analysed using framework analysis, which is suitable for mixed methods data in health research.[53 54]

Across the review period for this protocol, a small number of changes were made to our methods, namely: (1) for clarity, naming our process as AEBCD rather than EBCD; (2) reducing our AEBCD sample size from n=15 young people and n=15 professionals in each region to n=5/8 of each per region to optimise the experience of diverse young people in the workshops; (3) dropping purposive sampling in the implementation phase so that more settings will be eligible to test the resource and (4) removing the plan to conduct a system readiness assessment of settings prior to implementation in order to reduce the burden on participating settings.

## ETHICS AND DISSEMINATION

This study has received ethical approval from the UK National Health Service Health Research Authority (23/WM/0105 June 2023). Recruitment for the project will commence in summer 2023 and all study activities will be completed by August 2025. Organisations and participants will receive age-appropriate verbal and written information about the study, including as much transparency as possible about the workshops, who will be there, what is involved and how we are endeavouring to make the workshops psychologically safe. Participants will be reminded that they are free to take breaks in the workshop, to contribute in ways that are right for them, including just listening. Support for participants will be available in the workshops from trained clinicians and options for support beyond this will be made clear. Organisational consent for the implementation phase will be secured after the AEBCD stage. For any level of study participation, we will secure signed informed consent, with additional guardian consent for under 16s, and opt-out consent for the filmmaking components. All participating adolescents and organisations, and their data, will be fully anonymised in any study outputs.

Dissemination will be within this study first, via workshop 4 in the AEBCD series to the study participants and organisations, and then to the larger Attune study. Wider knowledge dissemination will be in partnerships with adolescents and participating organisations and will target academic, healthcare, education, social care, third sector and local government settings via knowledge exchange events, social media, executive reports, conference presentations and publications. We will share learning and insights about (1) conducting AEBCD on trauma projects with adolescents and UK public sector settings, using arts-based approaches on lived experience; (2) the concerns, priorities and opportunities adolescents and different sectors perceive for working in more trauma-informed ways; (3) the resource they produced and how it was implemented in diverse settings; (4) how resources for TIAs in diverse settings could be evaluated and (5) next steps for resource refinement and upscaling, if appropriate. Knowledge products will include a study website detailing the AEBCD process, the resource, evaluation and outcomes, and a youth-created film documenting the study process and organisational experiences of implementation.

## DISCUSSION

Trauma is so widespread and its effects are so significant, that is, it has been positioned as a significant public health issue. ACEs and trauma are highly predictive of poor outcomes in all areas of life and across lifespans, with adolescence being a time of particular vulnerability to the presentation of poor mental health in ACE-affected young people. Trauma-informed ways of working with young people strive to understand the ways that their ACEs may be shaping their needs and engagement with a service, with efforts to minimise retraumatisation. Although resources are emerging in the UK to support public sector settings to be trauma-informed, we are still at a preliminary stage of understanding about how they may be best implemented and sustained and whether they meet the needs of young people.

To our knowledge, this is the first AEBCD study working across different UK public sector settings to codesign a resource that could help professionals be more attuned to the needs of adolescents in their settings who have lived

through ACEs and/or trauma. Our study turns into action the learning from the arts-based participatory methods in the main Attune study, which captured the lived experience of diverse, ACE-affected adolescents in multiple UK regions. A number of strengths benefit the study. Involvement of different public sector settings allows us to identify commonalities in needs. We support collaboration between adolescents and professionals, inviting them to join the project together and work in dialogue across the AEDCD workshops. We address the need to better understand how resources for TIA can be implemented and evaluated in the public sector, driven by settings/services themselves and informed by NPT. This study will produce new insights on what adolescents and professionals want in a codesigned UK resource to improve their capacity to work in trauma-informed ways, and whether that resource is acceptable, feasible, efficacious. This contributes to the national agenda for developing TIAs for young people.

The study has some limitations. Recruitment of young people under 16 will be subjected to guardian consent; this may exclude young people who do not wish to or cannot secure that consent. The AEBCD process of three workshops may make project participation possible for stretched organisations who can only give limited time to research projects but means deep exploration and creativity may be compromised. Bringing diverse settings together may risk consensus-reaching on the type of resource to be created. The effects of trauma-informed organisational interventions can be slow to emerge and examine. Thus, in our time-limited project, it is an exploratory evaluation of their experiences implementing the resource to inform resource improvements, more effective implementation strategies and a more robust evaluation at a future point.

**Acknowledgements** The authors wish to thank each member of the wider ATTUNE research team who have contributed to the wider work of the project. The authors wish to thank the UKRI MRC for the funding and support. The authors also wish to acknowledge the support from the National Institute for Health and Care Research (NIHR) Applied Research Collaboration Oxford and Thames Valley at Oxford Health NHS Foundation Trust. The views expressed are those of the author(s) and not necessarily those of the NHS, the NIHR or the Department of Health and Social Care.

**Contributors** KB is the principal investigator for the ATTUNE study (ATTUNE— Department of Psychiatry (ox.ac.uk)) SH-J, KB and IB made substantial contributions to the conceptualisation and design of the study. IB and SH-J drafted the first version of the manuscript. IB, SH-J and KB reviewed the manuscript for intellectual content, approved the final version to be published and agreed to be accountable for all aspects of the work.

**Funding** This work was supported by the UKRI Medical Research Council. (MR/ W002183/1).

**Competing interests** None declared.

**Patient and public involvement** Patients and/or the public were involved in the design, or conduct, or reporting, or dissemination plans of this research. Refer to the Methods section for further details.

**Patient consent for publication** Not applicable.

**Provenance and peer review** Not commissioned; externally peer-reviewed.

**ORCID iDs**
Siobhan Hugh-Jones http://orcid.org/0000-0002-5307-1203
Isabelle Butcher http://orcid.org/0000-0003-2915-8269
Kamaldeep Bhui http://orcid.org/0000-0002-9205-2144

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
