## [Reviewer comments · BMJ Open]

ARTICLE DETAILS

TITLE (PROVISIONAL)	Co-design and evaluation of a youth-informed organisational tool to enhance trauma-informed practices in the UK public sector: a study protocol
AUTHORS	Hugh-Jones, Siobhan; Butcher, Isabelle; Bhui, Kamaldeep

VERSION 1 – REVIEW

REVIEWER	Jolley, Suzanne King's College London
REVIEW RETURNED	14-Sep-2023

GENERAL COMMENTS	The paper presents a protocol for a mixed methods evaluation of the creation and implementation of a youth-specific resource to promote trauma informed approaches in the public sector. The work sounds worthwhile and valuable. However, I find the protocol to read more like a grant application – a broad description of what will be done and aims, rather than the specifics of the work and clear statements of proposed practice against which the work can later be scrutinised. I cannot see that a specific template was used and this would be needed. Risks to the research are not discussed and thus not mitigated. Proof reading is needed. Some specific additional points follow. Introduction P3 para 2 – effect of absence of buffering – needs a reference Para 3 line 4 – adaptive following trauma but misconstrued – helpful to refer to overgeneralised learning here – behaviours are adaptive if further trauma is occurring, and tend to persist/spill over once trauma has stopped and into non-trauma situations Could a rationale be given for the 10-24 age range P7 Setting – the regions and their 'wider areas' do not tally P7 Participants – diverse or locally representative of population experiencing ACEs? Pros and cons of each? Later comment on diversity excludes gender identity and neurodivergence. How will selection proceed? Is the sample size adequate for the purpose? By what benchmark? P8 line 3 – provision of guardian consent – how will this impact participation for young people still residing in adverse contexts, and thus the representativeness of the study population? P8 para 1 and 2 – sounds a bit non-specific still for a protocol – exactly how will organisation and participants be identified and recruited? Para 3 – how will purposive sampling be conducted? Could more detail be given on the measurement framework, including its name in full and relevant references.
--

	P9/10 - Same for the ECBD workshop processes – at the moment it would be very hard to judge later whether the study had been conducted according to protocol or not. P10 – opt-in or out? I think this kind of distinction is important for a study protocol (and, I would have thought, for the ethical approval?) P11 – analysis – when will it be relevant to identify organisation type? Surely this information will be included or not? P12 suggests place data will also be retained
--	--

REVIEWER	Genç, Emel Bartın University, Psychology
REVIEW RETURNED	29-Sep-2023

GENERAL COMMENTS	The topic addressed by this paper is obviously an important one in developing trauma-informed practices in the public sector. However, there are certain places that need to be revised or addressed. First of all, I would make sure you ask at least 1-2 friends to read the paper thoroughly to make sure it is at a much better level, especially for minor language issues. Second, please explain the importance of working with youth groups since this protocol is developed for professionals who work with adolescents. Also, please explain why specifically, the authors selected the age range 10-24y? I think between 10- 24 y, there is a huge gap, and the developmental needs of this population would be different. I would like to see the content of the EBCD protocol to understand how the authors address the developmental differences of this population. Research question: I am not sure if this study design and results addressed all the research questions. It seems those questions belong to a different study. Method: the description of the method is vague and is missing details about the research procedures. It would be helpful to have a little more detail about EBCD and how it is developed. Also, the recruitment process is not made entirely clear. what were the procedures and inclusion criteria while recruiting participants? Please report a bit on the descriptives of the participants, how many of them will be minors? Are the researchers interviewing minors in this study?! Do they mention group sizes? More detail is needed here. Data Analysis: The authors could be more transparent about the process of data analysis. Lastly, the authors should discuss the potential contribution of their study to the literature and discuss the potential limitations. I hope that may suggestions could be helpful for the authors in improving their work.
---

VERSION 1 – AUTHOR RESPONSE

Reviewer 1

The paper presents a protocol for a mixed methods evaluation of the creation and implementation of a youth-

Thank you for your comments. Please accept my apologies but we are unsure of what 'template' refers to the the suggested revisions, e.g. template for EBCD or for a protocol paper. We do follow a template for EBCD, and cited Moser et al. (2022). For greater clarity, in our revised

specific resource to promote trauma informed approaches in the public sector. The work sounds worthwhile and valuable. However, I find the protocol to read more like a grant application – a broad description of what will be done and aims, rather than the specifics of the work and clear statements of proposed practice against which the work can later be scrutinised. I cannot see that a specific template was used and this would be needed.

P3 para 2 – effect of absence of buffering – needs a reference

Para 3 line 4 – helpful to refer to overgeneralised learning here.

Could a rationale be given for the 10-24 age range

P7 Setting – the regions and their 'wider areas' do not tally

P7 Participants – diverse or locally representative of population experiencing ACEs? Pros and cons of each? Later comment on diversity excludes gender identity and neurodivergence. How will selection proceed?

Is the sample size adequate for the purpose? By what benchmark?

manuscript Section 2.2, we have changed 'adapted EBCD' to accelerated EBCD (AEBCD) to indicate specifically our alignment to that method and provide citations. We have also reviewed several EBCD papers published in this journal and closely mirror their structuring (template) of the paper for journal consistency. EBCD also requires a degree of flexibility to ensure power relations are balanced in the process; and there is not a standardised way to do EBCD as it always needs to be built around participants. We have explained more on this need for flexibility and rationale.

Thank you for this, we have added a reference here.

This has been added, with a reference.

This is an excellent suggestion. We have explained (at the end of the Introduction) that this is the age range encouraged by the funder (UKRI) and is reflective of broader moves to consider the adolescent stage as starting earlier and finishing later than 13-19yrs. We provide a reference to this call in The Lancet Child and Adolescent Health.

We agree this was confusing and related to other Attune regional links. We have simplified this to three counties we are working in Cornwall, West Yorkshire and Kent.

We have clarified our recruitment process under Section 2.4.1, which is that participating organisations will be asked to identify suitable young people and approach them to explore interest in taking part.

We are interested in setting diversity but also in individual diversity in terms of place, ethnicity, gender, neurodiversity and LGBTQ+. We have stated in 2.4.1 that we will encourage participating organisations to consider active inclusion of typically underrepresented groups. In Attune, we operate on a basis of 'working with' rather than 'doing to' organisations, so that organisations feel respected as knowledgeable about their young people. Given this and our recruitment of a volunteer sample, we set no specific diversity targets.

I apologise for lack of clarity. The Point of Care Foundation suggests between 5-15 'service users' but no specified sample size for a professionals' group.

<https://www.pointofcarefoundation.org.uk/resource/experience-based-co-design-ebcd-toolkit/step-by-step-guide/7-recruiting-patients/>

Sample sizes vary widely in published EBCD and AEBCD studies, including in BMJ Open. For example, Bielinska et al. (2022) n=12 professionals in AEBCD;

Mooney et al. (2022) = 160 (15 - 20 from each local system); Brown et al. (2020) n= total sample of 40.

Our sample size decision was based on what would be a workable group size to capture some diversity but to allow effective group work during the co-design session, and where there would be at least the same number of young people as professionals. We decided this would be no more than the recommended n=15 (service user / young people), and more than 30 participants at each regional workshop. We have added a brief explanation of this to Section 2.4.1.

sP8 line 3 – provision of guardian consent – how will this impact participation for young people still residing in adverse contexts, and thus the representativeness of the study population?

We agree - it is possible that some young people who are still living in adverse contexts cannot ask for guardian consent are thus not represented in this study. We have added a line to this effect at the end of section 2.4.1. Of course, seeking research involvement may also be unethical for highly vulnerable young people, who may not be in sufficiently stable conditions. Requiring guardian consent is a concern in many sensitive studies with under 16s (e.g. on queer identities (Cwinn et al., 2021

<https://www.sciencedirect.com/science/article/pii/S1054139X20304572>). young people. We will reflect on representation of the final study sample in all outputs.

P8 para 1 and 2 – sounds a bit non-specific still for a protocol – exactly how will organisation and participants be identified and recruited?

Thank you for this comment. We have added to p9 “This may be through the Attune project network or by first approaches to local councils, local authorities, and / or third sector organisations.”

Para 3 – how will purposive sampling be conducted? Could more detail be given on the measurement framework, including its name in full and relevant references.

We have removed reference to purposive sampling. We have simplified our entry to the implementation phase and, from Section 2.4.1, have removed reference to the NASSS framework (Greenhalgh et al., 2017). We have stated that any organisations taking part in the EBCD stage can trial the resource in the implementation stage. This is to ensure there is return on participants’ time given to EBCD stage, and to optimise our learning from different settings. I do hope this is acceptable.

P9/10 - Same for the ECBD workshop processes – at the moment it would be very hard to judge later whether the study had been conducted according to protocol or not.

Thank you for this comment and suggestion. We have added more detail about the EBCD workshop activities in Table 1 and in Supplementary Materials. It is important for us not to over specify so that we can be responsive in workshops to the needs and capacities of particular young people and professionals who opt in. We now state this clearly in the protocol. For example, if we have a group of mostly 18-25y olds, we may opt to use an activity that is more challenging than would be possible with a young group, without changing the overall aim of the activity. We also wish to retain flexibility to respond to issues arising (eg related to triggering or the pace of discussion). Our workshop facilitators have a portfolio of alternative tasks that could be deployed responsively in each workshop.

P10 – opt-in or out? I think this kind of distinction is important for a study protocol (and, I would have thought, for the ethical approval?)

Once again thank you for this comment and suggestion. This is outlined in Section 3.0. We have added a sentence on p10 for extra clarity.

P11 – analysis – when will it be relevant to identify

Yes organisation type is recorded, but we avoid linked it to any quotations to limit identification. For clarity, we have now added to ‘Data

organisation type? Surely this information will be included or not? P12 suggests place data will also be retained

Analysis' that it may be relevant to identify the organisation type when knowledge of the setting type is required to comprehend the point / discussion. Yes, place (ie UK regions) will also be recorded.

We have removed reference to a live animator attending the workshops as this was deemed out of budget.

We have removed reference to Framework Analysis for workshop data as we do not have the personnel to complete this and we can deliver our main EBCD aims without it.

Reviewer 2

The topic addressed by this paper is obviously an important one in developing trauma-informed practices in the public sector. However, there are certain places that need to be revised or addressed. First of all, I would make sure you ask at least 1-2 friends to read the paper thoroughly to make sure it is at a much better level, especially for minor language issues.

Second, please explain the importance of working with youth groups since this protocol is developed for professionals who work with adolescents.

Also, please explain why specifically, the authors selected the age range 10-24y? I think between 10-24 y, there is a huge gap, and the developmental needs of this population would be different.

I would like to see the content of the EBCD protocol to understand how the authors address the developmental differences of this population.

Also, the recruitment process is not made entirely clear. what were the procedures and inclusion criteria while recruiting participants? Please report a bit on the descriptives of the participants, how many of them will be minors? Are the researchers interviewing

Thank you for your kind words. We have read this paper thoroughly and upon closer examination have been able to refine it, and so in doing so I hope the language has been improved.

At the end of the Introduction, we have added an explanation of the importance of working with young people.

Thank you for this comment. We have added (at the end of the Introduction) that this is the age range encouraged by the funder (UKRI) and is reflective of broader moves to consider the adolescent stage as starting earlier and finishing later than 13-19yrs. We provide a reference to this call in The Lancet Child and Adolescent Health.

I apologise this was not included in the original submission. We have given more detail in section 2.5.1 'AEBCD Workshop Series ' and in Table 1. We have also provided more detail in Supplementary Materials.

Thank you for these helpful suggestions. We have added more details on recruitment and sample size in Section 2.4.1

We had already outlined that our age range is 10-24y. We do not refer to 'interviewing minors', but young people from age 10y are eligible to take part in the EBCD workshops in the company of some participating professions from their linked organisation (e.g school, youth group). We cannot report more descriptive of the sample until we have recruited.

minors in this study?! Do they mention group sizes? More detail is needed here.

Data Analysis: The authors could be more transparent about the process of data analysis.

Lastly, the authors should discuss the potential contribution of their study to the literature and discuss the potential limitations.

This is a volunteer sample. The inclusion criteria are reported in section 2.4.1

We have added more detail under 'Data Analysis' for the co-design stage . We will not be subjecting the workshop recordings to more formal qualitative data analysis, as this removes the stakeholders from the process and also means the pace of design and implementation is no longer compatible with the intended schedule, whereby participants take the learning back to their organisations, knowledge diffusion, as EBCD progresses.

We did not have a section on this in the original submission as it is not a standard section for BMJ Open Protocols (see <https://bmjopen.bmj.com/pages/authors#protocol>) but we have now added this if the journal wishes to accept.

This study is novel, and builds upon the existing literature on adversities but also addresses the gap- little research has been conducted with young people utilizing EBCD.

This study furthermore will provide we believe resources for teachers and employers based on EBCD means privileging lived experience to generate novel insights and interventions, positioning this as valid new data and not dismissing it as subjective experiences; as doing so compounds epistemic injustice in research and in support.

VERSION 2 – REVIEW

REVIEWER	Jolley, Suzanne King's College London
REVIEW RETURNED	08-Dec-2023

GENERAL COMMENTS	Many thanks to the authors for their responses which have improved the manuscript. With regard to my first point about a template, I meant a framework for what must be included in a protocol for this kind of study, such that the authors can check that they have included relevant information - https://www.hra.nhs.uk/planning-and-improving-research/research-planning/protocol/ - is a generic example, the cited paper does not provide a framework of this kind. This remains an omission, in my view.
--

REVIEWER	Genç, Emel Bartın University, Psychology
REVIEW RETURNED	30-Dec-2023

GENERAL COMMENTS	Abstract: Page 2. Lines 23-40 (Methods and Analysis) were written like a research proposal, not a manuscript. Please delete the future tense (“will”) used in the sentences. Page 2, line 58. I didn't get “five” inside the parenthesis. Do the authors refer to five Strengths and Limitations? Another comment
---

	for this title: please write the Strengths and Limitations briefly and in one paragraph since this subtitle is under the abstract. Introduction: Page 4, line 52: “Trauma-informed approaches” What are these approaches? Please introduce them as well. Method: Page 7: The design section needs to be elaborated and clarified. For example, the authors mention “pre-test and post-test,” so if you use such a design, I expect the authors to mention the experimental study design, including treatment conditions. Also, this study design introduced in the section is not suitable for the first research question. Page 7. Line 33: “EBCD” what is it? Page 7, line 44: “AEBCD” What does that mean? Page 7, lines 40-60: please check the grammar of the paper. In some places, the authors use the present tense, but in some places, they use the future tense. Please be consistent throughout the paper. Page 9, Line 34-39: how would the authors assess if the participants “experienced trauma and/or ACE” Page 12. Line36-58: What method will be used for data analysis Page 13, line 34: “We will examine how the discovery phase impacted..” Is the discovery phase a part of workshop 1? I think before examining the discovery phase, it would be better if the authors explained the phases for each workshop. Page 16-18. I like Table 1, which is more informative than “Workshop” subtitles. The authors may elaborate on the in-text writings, based on the information on the table.
--	---

VERSION 2 – AUTHOR RESPONSE

Reviewer comments	Actions
Please note that the study dates in your Ethics and Dissemination section differ from those provided in your letter. Can you please clarify whether this should be updated in the manuscript?	I apologise for any confusion caused. Recruitment for the project will commence in summer 2023 and all study activity will be completed by August 2025
*If relevant, we advise including a paragraph with details about late changes to the protocol in the Methods section for transparency. We note that some changes seem to have been made following reviewer 1's comments, e.g. the numbers of participants in the study and the use of purposive sampling.	We have now added a short paragraph to the end of the Methods section on late changes to the protocol.
Comments to the Author:	
Many thanks to the authors for their responses which have improved the manuscript. With regard to my first point about a template, I meant a framework for what must be included in a protocol for this kind of study, such that the authors can check that they have included relevant information -	We thank the reviewer for encouraging inclusion of a protocol template. The Attune project does have a full protocol (that was submitted to HRA ethics) and this followed the key headings provided in the HRA template including, background, aim, study

https://www.hra.nhs.uk/planning-and-improving-research/research-planning/protocol/ - is a generic example, the cited paper does not provide a framework of this kind. This remains an omission, in my view.	design, sample, recruitment plan, ethical considerations, data analysis plan. It should be noted that there is currently no standardised protocol templates for studies that include creative methodologies, so when submitting our HRA application we also looked to templates and information provided by the King's Fund and Medical Research Council to ensure that the protocol was covered in sufficient detail. We have also noted that BMJ Open does not require that study protocol papers include such a protocol template. A review of several recent protocol papers in BMJ Open shows this is not standard practice. We feel there is sufficient detail in our paper for transparency and reporting checks.
Reviewer: 2	
Dr. Emel Genç, Bartın University	
Comments to the Author:	
Abstract:	
Page 2. Lines 23-40 (Methods and Analysis) were written like a research proposal, not a manuscript. Please delete the future tense ("will") used in the sentences.	As this is a research protocol, setting out plans for the conduct of our study, the future tense is appropriate. We have checked the manuscript again to ensure tenses are meaningful.
Page 2, line 58. I didn't get "five" inside the parenthesis. Do the authors refer to five Strengths and Limitations? Another comment for this title: please write the Strengths and Limitations briefly and in one paragraph since this subtitle is under the abstract.	Five refers to the number of strengths and limitations; we apologise for confusion and have deleted this. The format for BMJ Open protocols is to present Strengths and Limitations as bullet points.
Introduction:	
Page 4, line 52: "Trauma-informed approaches" What are these approaches? Please introduce them as well.	Thank you for this question. Trauma-informed approaches aim to improve the capacity of professionals and services to

	respond helpfully to people who may have experienced trauma. This has now been added to the protocol.
Method:	
Page 7: The design section needs to be elaborated and clarified. For example, the authors mention “pre-test and post-test,” so if you use such a design, I expect the authors to mention the experimental study design, including treatment conditions. Also, this study design introduced in the section is not suitable for the first research question.	This is a co-design and evaluation study (with before and after measures), and as an early-stage study, does not use a comparison group or experimental design. We therefore do not draw on the term ‘treatment groups’, and additionally, as our work is situated in public mental health and prevention agenda. Our first line of the Study Design section is, we feel, suitable to our first research question as we will be addressing this in co-design.
Page 7. Line 33: “EBCD” what is it?	This is Experience Based Co Design mentioned which we have now clarified.
Page 7, line 44: “AEBCD” What does that mean?	This is Accelerated Experience Based Co Design, as stated in the Introduction section and also clarified again in the Methods.
Page 7, lines 40-60: please check the grammar of the paper. In some places, the authors use the present tense, but in some places, they use the future tense. Please be consistent throughout the paper.	At times we refer to future plans, we use future tense. At other times, where appropriate, we use the present tense (e.g. to refer to the implications of a study).
Page 9, Line 34-39: how would the authors assess if the participants “experienced trauma and/or ACE”	We have now added the following for additional clarity in section 2.4.1: That recruited young people will have experienced trauma and / or ACEs will be determined via three means: (i) knowledge that the recruiting setting has about the young person whom they deem suitable to approach for study participation; (ii) our recruitment information to young people will include experience of trauma / ACEs as an eligibility criterion; and (iii) via a survey administered as part of the larger Attune survey that includes items about experiences of ACEs and trauma.
Page 12. Line36-58: What method will be used for data analysis	In Section 2.5.1, we had detailed our plans for data analysis. This study does not require

	an in-depth qualitative analysis, since it is already drawing on the outcomes of such an in-depth analysis conducted as part of the larger Attune project (as explained in the section on AEBCD). Our approach to data handling to drive to co-design process is documented and is comparable with similar co-design studies.
Page 13, line 34: “We will examine how the discovery phase impacted..” Is the discovery phase a part of workshop 1? I think before examining the discovery phase, it would be better if the authors explained the phases for each workshop.	We politely direct the reviewer that we had already explained the discovery phase of AEBCD in section 2.2.
Page 16-18. I like Table 1, which is more informative than “Workshop” subtitles. The authors may elaborate on the in-text writings, based on the information on the table.	We thank the reviewer for this suggestion. We considered adding more detail to Table 1, but felt that, in conjunction with the narrative in the main manuscript, and the supplementary materials, that sufficient detail about the workshops is already presented.